# Stigma and social support and their impact on quality of life and self-esteem among women with endometriosis in Latin-America and the Caribbean

Yatzmeli Matías-González[1], Astrid Sánchez-Galarza[1], Ernesto Rosario-Hernández[1], Idhaliz Flores-Caldera[2,3], Eliut Rivera-Segarra[1] *

1 School of Behavioral and Brain Sciences, Ponce Health Sciences University, Ponce, Puerto Rico,
2 Department of Basic Sciences, Ponce Health Sciences University, Ponce, Puerto Rico, 3 Department of Ob-Gyn, Ponce Health Sciences University, Ponce, Puerto Rico

* elrivera@psm.edu

## Abstract

Endometriosis is a complex disease affecting approximately 5–10% individuals worldwide. Prevalence rates in Latin-America and the Caribbean are largely unknown, with published data only from Puerto Rico and Chile. Pain associated with endometriosis negatively affects patients' health and quality of life. However, there is a need to better understand the role played by psychosocial variables such as stigma and social support in diverse cultural contexts. The purpose of this study was to examine the mediating role of endometriosis related stigma (endo-stigma) and the moderating role of social support on the endometriosis QoL and self-esteem among women with endometriosis from Latin America and the Caribbean. A cross-sectional design with online survey techniques was implemented. A total of 169 self-identified cisgender women with endometriosis from 14 Latin-American and Caribbean countries participated in the study. We used partial least squares structural equation modeling (PLS-SEM) to examine the study's hypotheses. Incapacitating pain was positively and significantly related to endometriosis QoL as measured by the EHP-5 (b = .266, p < .01). Endo-stigma was positively and significantly related to endometriosis QoL (b = .340, p< .01) and self-esteem (b = .297, p< .01). In addition, endo-stigma mediated the relationship between incapacitating pain and self-esteem (IE = .073, p = .018). Finally, social support moderated the relationship between stigma stress and endometriosis QoL (b = .060, p = .039). Findings suggest stigma could be one of the mechanisms through which the relationship between incapacitating pain and self-esteem among Latin American and Caribbean women with endometriosis could be partially explained. Furthermore, women who scored high in the need for social support and stigma stress also showed worst endometriosis QoL. These results point towards the need to develop tailored interventions targeting these factors in order to foster a better QoL and wellbeing for this population in the context of Latin America and the Caribbean.

**Data Availability Statement:** The manuscript presents the data used in this study. Although data

is not publicly available in order to comply with the Institutional Review Board (IRB) dataset requests will be evaluated on a case by case basis by the IRB chair to ensure it is a reasonable request. People interested in accessing the data set can directly contact the IRB chair via email at: scarlo@psm.edu.

**Funding:** This publication was supported by the Ponce Research Institute, the School of Behavioral and Brain Sciences and the National Institute on Minority Health and Health Disparities under award U54MD007579. E R-S is also supported by the National Institute of Mental Health under award R34MH120179 and the National Institute on Minority Health and Health Disparities under award U54MD007579. I F-C is supported by the National Institute of Child Health and Human Development under award R21HD098481. The content is solely the authors' responsibility and does not necessarily represent the official views of the National Institutes of Health. The funders had no role in study design, data collection and analysis, decision to publish, or preparation of the manuscript. The funders had no role in study design, data collection and analysis, decision to publish, or preparation of the manuscript.

**Competing interests:** The authors have declared that no competing interests exists.

## Introduction

Endometriosis is a complex but common inflammatory condition affecting 5 to 10% of individuals worldwide [1]. Endometriosis can have a disabling physical and psychological impact. On the one hand, physical symptoms often include, but are not limited to dysmenorrhea, chronic pelvic pain, dyspareunia, fatigue, back and leg pain, gastrointestinal and urinary symptoms, and even infertility [1]. On the other, psychosocial outcomes often include reduced quality of life (QoL), poor self-esteem and even depressive and anxiety symptomatology [2, 3]. Furthermore, some evidence suggests that there might be a relationship between endometriosis physical symptoms (i.e., pelvic pain) and psychological symptomatology (i.e., anxiety) [2]. Unfortunately, most research to date comes from Europe and the United States (US), leaving people from Latin American and the Caribbean poorly represented in endometriosis research and databases. Thus, the impact of the condition and available treatments among people in this world region is still largely unknown with only a few studies from Puerto Rico, Chile, and Brazil [4–7]. In Puerto Rico, the diagnosis and treatment of endometriosis is often provided by community-based general gynecologists who do not always have specialized training in endometriosis. Globally, other countries and regions of the world also lack of comprehensive, multilevel interventions and integrated care for endometriosis patients [8]. Moreover, most research to date addresses endometriosis from biological and epidemiological frameworks, neglecting to understand the role of psychosocial and cultural variables in the experience of endometriosis.

Endometriosis-related stigma (endo-stigma) has been recently identified as a key under-researched psychosocial variable in need of further attention in endometriosis research [9, 10]. Stigma is a complex social process through which human differences are labeled and rejected. Thus, it has been identified as a social determinant of health that impacts the access to treatment, self-esteem, and social support systems of people living with chronic illnesses, such as endometriosis [11]. The available studies on endo-stigma have consistently found that it has a detrimental impact on people living with endometriosis by delaying diagnosis and damaging interpersonal relationships among families' members, intimate partners, and healthcare personnel [10, 12, 13]. Recent research found the anticipation of endo-stigma, due to the disruption of endometriosis symptoms, to be highly prevalent among women living with endometriosis in the US [14]. Furthermore, anticipated endo-stigma was found to be even higher among those born outside the continental US and with other marginalized intersecting identities (i.e., immigrants, sexual and gender minorities and Black and Latinas). However, despite these recent findings, research on endo-stigma continues to be an under-studied topic with only a few published qualitative studies and one quantitative study that the authors are aware of to date [10, 12–14]. Thus, there is an urgent need to better understand the role of endo-stigma in the experience of endometriosis worldwide, but especially among underrepresented and historically excluded communities and world regions such as Latin America and the Caribbean.

Therefore, the general objective of this study was to understand the role of endo-stigma on the quality of life among a sample of people living with endometriosis in Latin America and the Caribbean. To achieve this, we specifically examined: (1) the direct effects of incapacitating pain, endo-stigma, stigma stress and social support on endometriosis QoL and self-esteem, (2) the moderating role of need for social support and, (3) the mediating role of stigma on the endometriosis QoL and self-esteem (see research model in Fig 1).

## Methods

We implemented a cross-sectional design with an anonymous self-administered online survey technique using the online platform Research Electronic Data Capture (REDCap) hosted at

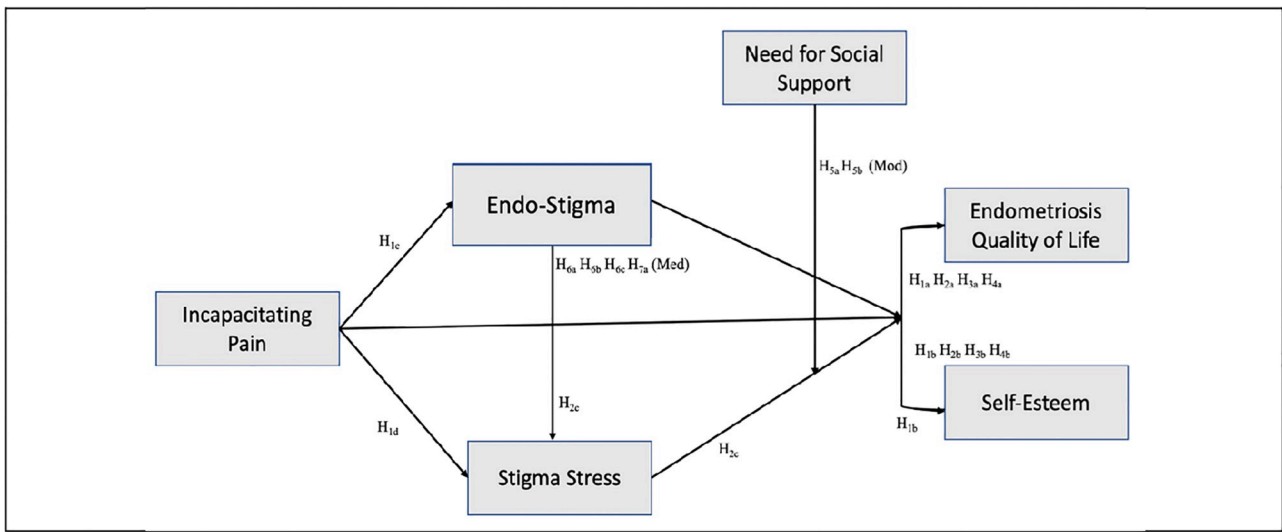

**Fig 1. Proposed research model.**

Ponce Health Sciences University. REDCap is a secure, web-based application designed to support data capture for research studies [15]. We used a non-probability sampling strategy. Data was collected between December 2019 and May 2020. Promotional information about the study and an electronic link to the survey was shared through webpage email listings and social media platforms (i.e., Facebook) of active endometriosis patients advocacy associations in Latin America and the Caribbean such as *Fundación Puertorriqueña de Pacientes con Endometriosis* (ENDOPR), *Asociación Colombiana de Endometriosis e Infertilidad (ASOCOEN)*, *Asociación Endometriosis Panamá (AENPA)*, *Endometriosis República Dominicana*, *Endometriosis México*, *Asociación Costarricense Endometriosis (AENDOCR)* and, *Endo Team Perú*.

## Ethics statement

This study obtained the Ponce Health Sciences University Institutional Review Board (IRB) approval (Protocol #1904010918). Potential participants accessed the consent form via the shared online link. The link was shared through social media and among endometriosis advocacy associations. Once the consent forma was completed, they were able to access the online survey.

## Reflexivity statement

All authors are Puerto Ricans living in Puerto Rico and some are people with lived experience of endometriosis. This was an online study, and thus shared widely in the web. Our study was conducted from Puerto Rico and shared with patient advocacy associations in Puerto Rico (a Caribbean country) and other Latin American and Caribbean countries which whom we have partnered before. We do not aim to speak for each individual country, but rather, examine broadly the region of Latin-America and the Caribbean.

## Participants

A total of 169 self-identified cisgender women diagnosed with endometriosis from 14 Latin American and Caribbean countries (Puerto Rico, Panama, Costa Rica, Argentina, Dominican Republic, Colombia, Venezuela, and Mexico) participated in the study. Most of our study

**Table 1. Sociodemographic data.**

| Variable | f | % | Variable | f | % |
|---|---|---|---|---|---|
| **Marital Status** | | | **Incapacitating Pain** | | |
| Single | 60 | 35.5 | Yes | 137 | 81.1 |
| Married | 65 | 38.5 | No | 24 | 14.2 |
| Living Together | 29 | 17.2 | **Pain While Menstruating** | | |
| Separated | 3 | 1.8 | Always | 116 | 68.6 |
| Divorced | 11 | 6.5 | Usually | 44 | 26.0 |
| Widowed | 1 | 0.6 | Rarely | 8 | 4.7 |
| **Education** | | | Never | 1 | 0.6 |
| ≤ High School | 7 | 4.1 | **Pain During Sex** | | |
| Undergraduate | 66 | 39.0 | Always | 31 | 18.3 |
| BA | 42 | 24.9 | Usually | 81 | 47.9 |
| Some Graduate | 17 | 10.1 | Rarely | 32 | 18.9 |
| MA | 29 | 17.2 | Never | 25 | 14.8 |
| Some Doctorate | 1 | 0.6 | | Mean | SD |
| Doctorate | 7 | 4.1 | **Age** | 33.15 | 7.44 |

Note: n = 169; SD = Standard Deviation.

participants were from Puerto Rico (N = 51). The inclusion criteria for this study were: (1) being 21 years of age or older, (2) being diagnosed with endometriosis via self-reported surgery or laparoscopy, (3) identify as a Latin American or Latinx and (4) use Spanish as their primary language of communication. The mean average age of study participants was 33.15 SD± 7.44, 38.5% were married, and 39.0% had at least some college-level education (see Table 1). In terms of pain, 81.1% of the participants reported experiencing incapacitating pain (pain that prevents them from doing daily tasks), 68.6% always experience pain during their periods, and 66.2% of our sample reported to usually experience pain during sex (see Table 1).

## Measures

**Demographic data questionnaire.** This questionnaire developed by the team, collected self-reported information on the sociodemographic data of the participants, such as: age, marital status, education, as well as endometriosis diagnosis and symptoms.

**Stigma Scale for Endometriosis (SSE).** Endo-stigma was measured using an adapted version of the original 8-item Stigma Scale for Chronic Illness (SSCI) developed by Molina et al. (2013) to specifically reflect endometriosis instead of chronic illness generally (by susbstituting "chronic illness" with "endometriosis") [16]. The SSCI was translated to Spanish by two native Spanish-speaking authors (YM, AS). A third translator (ERS) compared the translated version with the original version and search for inconsistencies. After this process, the translation was considered final when no differences or inconsistencies were found. The original and adapted version of the scales measure multiple aspects of stigma, such as internalized and enacted stigma. The SSCI has shown high internal consistency and validity to measure stigma in people living with chronic illnesses [16, 17]. The scale consists of Likert rated items ranging from Never (1) to Always (4). High scores in the adapted version of the SSCI, represents the presence of endometriosis related stigma.

**Stigma Stress Scale (CogApp).** The Stigma Stress Scale is an eight-item scale that examine the cognitive evaluation of stigma as a stressor [18]. It is rated using a Likert format from 1 to 7 points: (1) Strongly disagree to (7) Strongly agree. An adapted version of the questionnaire

was implemented, to specify and reflect stress specifically due to endo-stigma. High scores in this scale represents higher stress due to endo-stigma. The questionnaire was translated to Spanish using the same strategy described above.

**Social support questionnaire.** Need for social support was measured using and adapted version of the Social Support Questionnaire. This instrument was developed for the Medical Outcome Study (MOS) conducted at the RAND corporation, for patients with preventable and treatable conditions. This questionnaire was translated and validated for Spanish speaking populations in primary care settings [19]. This questionnaire measures the need for social support in four different domains: emotional, instrumental, positive social interaction, and affective interactions [20]. It is a scale that contains 19 items, rated with a Likert scale from "1" to "5" points: (1) Never, (2) Rarely, (3) Sometimes, (4) Often, and (5) Always. High scores correspond to the need for social support. Given that the construct of need for social support has four dimensions, which tends to be quite complex, we decided to use an adapted Social Support questionnaire as a high order model or hierarchical component model (HCM), as it is usually called in the context of PLS-SEM [21], to simplify the current research model. This reduces the number of relationships in the structural model, making the PLS path model more parsimonious and easier to grasp [22]; that is, instead of working with four dimensions, we only worked with one HCM, need for social support. Moreover, since the four dimensions of the MOS questionnaire tend be highly correlated (e.g., [23], the use of HCM help to deal with collinearity issues [22].

**Short Form Endometriosis Health Profile (EHP-5).** The EHP-5 is an instrument that measures domains of endometriosis related QoL: illness, physical ability, independent living, psychological state, and social interaction. It consists of two questions that assess how much endometriosis symptoms interfere with work and daily activities during the past 4 weeks with a Likert scale from 0 to 4 points: (0) Never, (1) Rarely, (2) Sometimes, (3) Often, and (4) Always. Higher scores represent poor endometriosis QoL. This instrument was developed in United Kingdom to measure the health status of women with endometriosis [24] and has been adapted for Spanish speaking populations in a sample of Puerto Rican women living with endometriosis [25]. Psychometric properties of the instrument reported a high internal consistency with a Cronbach's alpha between the range of .80 to .95.

**Rosenberg Self-Esteem Scale.** This is a self-report scale that evaluates an individual's self-esteem and how someone feels about themselves [26]. It consists of 10 items rated with a Likert scale of 4 points: (1) Strongly agree, (2) Agree, (3) Disagree, and (4) Strongly disagree. Higher values in this scale corresponds to higher perception of self-esteem. Lower scores represent significant self-esteem difficulties in our study subjects. Therefore, it has a range of total scores between 10 and 40. The range is one-dimensional both in the original version and in the Spanish version. We administered the Spanish version available in the literature adapted and validated for Spanish speaking populations [27]. The scale generally has high internal consistency and validity: test-retest correlations are typically in the range of .82 to .88, and Cronbach's alpha for various samples is in the range of .77 to .88 [26, 28].

## Data analysis

Descriptive statistics (means, standard deviation, frequencies, and normality) were conducted using the Statistical Package for Social Science (SPSS) version 27. We used partial least squares structural equation modeling (PLS-SEM) with the Smart-PLS 3.2.4. program to examine the study's objectives [29], which is a two-phase process. First, we assessed the psychometric properties of the measurement model by examining convergent, divergent, and reliability of the measurement instruments. If the measurement model is established, we move to the second

phase, which is the structural model. In the structural model we estimate the parameters of the structural model to test the study hypotheses (relationship between the multiple constructs in the model). PLS-SEM enables the simultaneous analysis of up to 200 indicator variables, allowing the examination of multiple mediator variables simultaneously among latent predictor variables indicators. Following Chin's [29] suggestion, it is important to mention the three reasons for its use in the present study. Firstly, PLS-SEM has soft distributional assumptions and given that the Kolmogorok-Smirnov and Shapiro-Wilks tests were significant, it suggested that scores and data were not distributed normally. Secondly, the exploratory nature of the current study [30–32], designed to examine the endometriosis QoL and self-esteem of those affected by endometriosis and how pain and stigma impact them. Lastly, the high model complexity of the study justifies the use of PLS-SEM because the model tested has multiple mediator variables [31, 32].

## Results

### The measurement models

The data indicates that the measures of endo-stigma, stigma stress, self-esteem and need for social support are robust in terms of their internal consistency reliability as indexed by Cronbach's alpha and composite reliability (see Table 2). All the Cronbach's alphas and the composite reliabilities of the different measures range from .77 to .97, which exceed the recommended threshold value of .70 [22]. In addition, consistent with the guidelines of Fornell and Larcker [33], the average variance extracted (AVE) for each measure exceeds .50, which is an indication of the convergent validity of the measures. Also, most outer loadings reached the threshold of .70, as indicated by Hair and colleagues [22]. It is important to mention that two outer loadings of the need for social support indicators are presented, since it was used as a second-order construct (MOS) and its respective subscale. Also, Table 2 shows that the results obtained with the surveys used in this study, MOS, SSE, and EHP-5, are valid and reliable as the composite reliability scores were over 0.70. In general, taking account 4 different dimensions measured using MOS we observed that need for social support modifies the stigma experienced with endometriosis. In addition, we studied two other variables that can potentially modify this relationship: self-esteem and quality of life.

The elements in the matrix diagonals, representing the square roots of the AVE, are greater in all cases than the off-diagonal elements in their corresponding row and column, supporting the discriminant validity of the scales (see Table 3 above the matrix diagonals). In terms of establishing the discriminant validity of the measures in the model, Henseler et al. (2015) propose assessing the heterotrait-monotrait ratio (HTMT) of the latent construct's correlations. The HTMT approach is an estimate of what the true correlation between two constructs would be if they were perfectly measure. A correlation between to constructs close to one indicates a lack of discriminant validity. Therefore, a threshold value of .90 is suggested if the path model includes constructs that are conceptually very similar. Also, because the HTMT can serve as the basic statistical discriminant validity test, the use of bootstrapping technique is recommended to derive a bootstrap with a 95% confidence interval with 5,000 random subsamples [34]. Thus, a confidence interval containing the value of one indicates a lack of discriminant validity. Because the HTMT-based assessment using confidence interval relies on inferential statistics, one should primarily rely on this criterion. In this study, none of the correlations between the constructs in the bootstrapping 95% confidence interval included the value of one; therefore, this suggests that the constructs are empirically distinct (see Table 3 below the matrix diagonals).

**Table 2. Outer loadings, average variance extracted (AVE), and reliability using Cronbach's alpha and composite reliability.**

| Latent Construct | Item | Outer Loading | AVE | Cronbach's Alpha | Composite Reliability |
|---|---|---|---|---|---|
| MOS-Need for Social Support | | MOS/Subscale | .66 | .97 | .97 |
| MOS-Emotional Support | MOS-3 | .74 / .82 | | | |
| | MOS-4 | .77 / .83 | | | |
| | MOS-8 | .77 / .80 | | | |
| | MOS-9 | .87 / .92 | | | |
| | MOS-13 | .86 / .88 | | | |
| | MOS-16 | .86 / .89 | | | |
| | MOS-17 | .89 / .90 | | | |
| | MOS-19 | .86 / .87 | | | |
| MOS-Instrumental Support | MOS-5 | .68 / .81 | .75 | .83 | .90 |
| | MOS-12 | .66 / .90 | | | |
| | MOS-15 | .63 / .88 | | | |
| MOS-Social Relationship | MOS-7 | .85 / .92 | .86 | .95 | .96 |
| | MOS-11 | .89 / .94 | | | |
| | MOS-14 | .85 / .90 | | | |
| | MOS-18 | .87 / .96 | | | |
| MOS-Affective Support | MOS-6 | .84 / .92 | .84 | .91 | .94 |
| | MOS-10 | .83 / .92 | | | |
| | MOS-20 | .84 / .91 | | | |
| Stigma Scale for Endometriosis | SSE-2 | .73 | .58 | .90 | .92 |
| | SSE-8 | .77 | | | |
| | SSE-9 | .75 | | | |
| | SSE-14 | .84 | | | |
| | SSE-15 | .83 | | | |
| | SSE-16 | .77 | | | |
| | SSE-18 | .63 | | | |
| | SSE-22 | .77 | | | |
| Stigma Stress | SS-1 | .79 | .54 | .81 | .87 |
| | SS-2 | .83 | | | |
| | SS-3 | .85 | | | |
| | SS-4 | .86 | | | |
| | SS-5 | .44 | | | |
| | SS-6 | .48 | | | |
| Self-Esteem | SE-1 | .62 | .53 | .77 | .85 |
| | SE-5 | .60 | | | |
| | SE-6 | .81 | | | |
| | SS-9 | .84 | | | |
| | SS-10 | .74 | | | |
| Endometriosis QoL | EHP5-1a | .71 | .56 | .81 | .87 |
| | EHP5-1b | .82 | | | |
| | EHP5-1c | .78 | | | |
| | EHP5-1d | .70 | | | |
| | EHP5-1e | .73 | | | |

## The structural model

After measures were tested for validity, the structural model in Fig 1 was tested. To examine the structural model [22], we initially checked for collinearity issues and examined the variance

**Table 3. Correlation matrix of latent variables using the fornell-larcker criterion and correlation proportion of heterotrait-monotrait (HTMT).**

| Latent Variable | NSS | SSE | SS | SE | EqoL |
|---|---|---|---|---|---|
| Need for Social Support (NSS) | (.81) | -.30 | -.37 | -.35 | -.31 |
| Stigma Scale for Endometriosis (SSE) | .31 [.174; .476] | (.76) | .56 | .46 | .52 |
| Stigma Stress (SS) | .42 [.272; .577] | .63 [.525; .725] | (.73) | .42 | .42 |
| Self-Esteem (SE) | .40 [.249; .551] | .54 [.394; .677] | .55 [.385; .701] | (.73) | .33 |
| Endometriosis QoL (EqoL) | .35 [.233; .485] | .59 [.468; .705] | .51 [.340; .679] | .43 [.317; .595] | (.75) |

Note: Elements in the correlation matrix diagonals within parenthesis represent the square roots of the AVE; elements above the diagonals represent the correlations between latent constructs; elements below the diagonals in the brackets are the confidence intervals of .90 for the HTMT's criteria correlations.

inflation factor (VIF) value of all sets of predictor constructs in the model. VIF values fluctuated between 1.011 and 1.582, within the threshold range of 0.20 and 5.00; therefore, collinearity among predictor constructs is not a critical issue in this structural model (see Table 3).

In addition, Table 4 represents the relationship among the constructs and latent variables and shows the $R^2$ values of endometriosis QoL (.399), self-esteem (.282), stigma stress (.317), and endo-stigma (.060), explaining 39.9%, 28.2%, 31.7% and 6.0% of the variance, respectively. Falk and Miller [35] suggest a value of .10 for an R-squared as minimum satisfactory level, therefore all endogenous latent variables possess the threshold level of R-squared values, except endo-stigma. Also, all $Q^2$ values of endometriosis QoL, self-esteem, stigma stress, and endo-stigma, are above zero (.204, .129, .158 & .031, respectively), providing support of the model's predictive relevance regarding the endogenous latent variables. The effects sizes for incapacitating pain achieved $f^2$ values of .064, .004, .001, & .110 on endometriosis QoL, self-esteem, endo-stigma, and stigma stress, respectively, which only exceeds the minimum threshold of .02 on endometriosis QoL and endo-stigma [36], while effect sizes for need for social support exceed the minimum threshold on all endogenous variables (see Table 4). Regarding the effect sizes for the interaction of stigma stress and need for social support on endometriosis QoL, this value exceeds, and it is considered a large effect for an interaction [37]. In general, this model explains 39.9% of the variance of the endometriosis QoL and 28.2% of the self-esteem variance.

In terms of the direct effects results (see Table 5), incapacitating pain had positive and significant relations to endometriosis QoL (b = .266, p < .001) and endo-stigma (b = .246, p < .001). On the other hand, endo-stigma had significant and positive relationship to endometriosis QoL (b = 339, p = < .001), self-esteem (b = .2979, p < .001), and stigma stress (b = .548, p < .001). Meanwhile, stigma stress did not have a significant relationship to endometriosis QoL (b = .115, p = .203) and self-esteem (b = .167, p = .105). Finally, need for social support had

**Table 4. Structural model results.**

| Latent Variable | $R^2$ | $R^2$ Adj. | Effect Size ($f^2$) | | | | $Q^2$ | VIF |
|---|---|---|---|---|---|---|---|---|
| | | | SSE | SS | SE | EqoL | | |
| Incapacitating Pain | | | .064 | .004 | .001 | .110 | | 1.068 |
| Need for Social Support (NSS) | | | | | .046 | .030 | | 1.180 |
| SS * NSS | | | | | .005 | .038 | | 1.011 |
| Stigma for Endometriosis (SSE) | .060 | .055 | | | | | .031 | 1.523 |
| Stigma Stress (SS) | .317 | .308 | | | | | .158 | 1.582 |
| Self-Esteem (SE) | .282 | .260 | | | | | .129 | |
| Endometriosis QoL (EqoL) | .399 | .380 | | | | | .204 | |

**Table 5. Hypotheses, results, and conclusions of direct and moderating effects.**

| Hypothesis (Path) | | beta | SE | t-value | p-value | CIBC | | Decision |
|---|---|---|---|---|---|---|---|---|
| | | | | | | 2.50% | 97.50% | |
| Hypothesis 1 | | | | | | | | |
| H$_{1a}$: | IP → EqoL | **.266**$^*$ | .065 | 4.115 | < .001 | .135 | .386 | Supported |
| H$_{1b}$: | IP → SE | .025 | .071 | 0.353 | .724 | -.122 | .155 | Not Supported |
| H$_{1c}$: | IP → SSE | **.246**$^*$ | .071 | 3.48 | .001 | .095 | .369 | Supported |
| H$_{1d}$: | IP → SS | .051 | .059 | 0.872 | .383 | -.066 | .166 | Not Supported |
| Hypothesis 2 | | | | | | | | |
| H$_{2a}$: | SSE → EqoL | **.339**$^*$ | .079 | 4.292 | < .001 | .180 | .492 | Supported |
| H$_{2b}$: | SSE → SE | **.297**$^*$ | .083 | 3.586 | < .001 | .114 | .449 | Supported |
| H$_{2c}$: | SSE → SS | **.548**$^*$ | .050 | 10.971 | < .001 | .435 | .635 | Supported |
| Hypothesis 3 | | | | | | | | |
| H$_{3a}$: | SS → EqoL | .115 | .090 | 1.273 | .203 | -.07 | .279 | Not Supported |
| H$_{3b}$: | SS → SE | .167 | .103 | 1.619 | .105 | -.038 | .367 | Not Supported |
| Hypothesis 4 | | | | | | | | |
| H$_{4a}$: | NSS → EqoL | **-.146**$^*$ | .067 | 2.164 | .031 | -.271 | -.006 | Supported |
| H$_{4b}$: | NSS → SE | **-.197**$^*$ | .084 | 2.351 | .019 | -.350 | -.025 | Supported |
| Hypothesis 4 (Mod) | | | | | | | | |
| H$_{5a}$: | SS * NSS → EqoL | **.152**$^*$ | .055 | 2.772 | .006 | .050 | .263 | Supported |
| H$_{5b}$: | SS * NSS → SE | .060 | .068 | 0.876 | .381 | -.073 | .195 | Not Supported |

Note: SE = Standard Error, CIBC = Confidence Interval Bias Corrected, IP = Incapacitating Pain, EqoL = Endometriosis QoL, SE = Self-Esteem, SSE = Stigma Scale for Endometriosis, SS = Stigma Stress, NSS = Need for Social Support.

significant and negative relationship to endometriosis QoL (b = -.206, p = .031) and self-esteem (b = -197, p = .019. Regarding the moderating effect of need for social support, it only moderated the relationship between stigma stress and endometriosis QoL (b = .152, p = .006). This moderating effect suggests that in this sample, those scoring high on stigma stress and the need for social support tend to present worse endometriosis QoL (see Fig 2).

Table 5 summarizes the moderating effects of the variables. Significant interactions included SS and NSS. This supports the idea that those with social support have better quality of life, and that those with worse stigma stress had worse endometriosis QoL.

In terms of the mediating role of endo-stigma (Table 6), it partially mediated the relationship between incapacitating pain and endometriosis QoL (IE = .083). Because the direct effect and indirect effect were significant and point in the same direction, it is considered as complementary mediation. However, endo-stigma completely mediated the relationship between incapacitating pain and self-esteem (IE = .073) and stigma stress (IE = .135). Because both indirect effects were significant while direct effects were not, this is considered as indirect effect only [38].

## Discussion

This study quantified and examined the relationship between incapacitating pain, stigma, and need for social support in the endometriosis QoL and self-esteem of women living with endometriosis in Latin American and the Caribbean. We also examined the moderating role of need for social support and the mediating role of endo-stigma on the endometriosis QoL and self-esteem of patients. To the best of our knowledge, this is the first cross-sectional and quantitative study that has examined endo-stigma in patients from this world region.

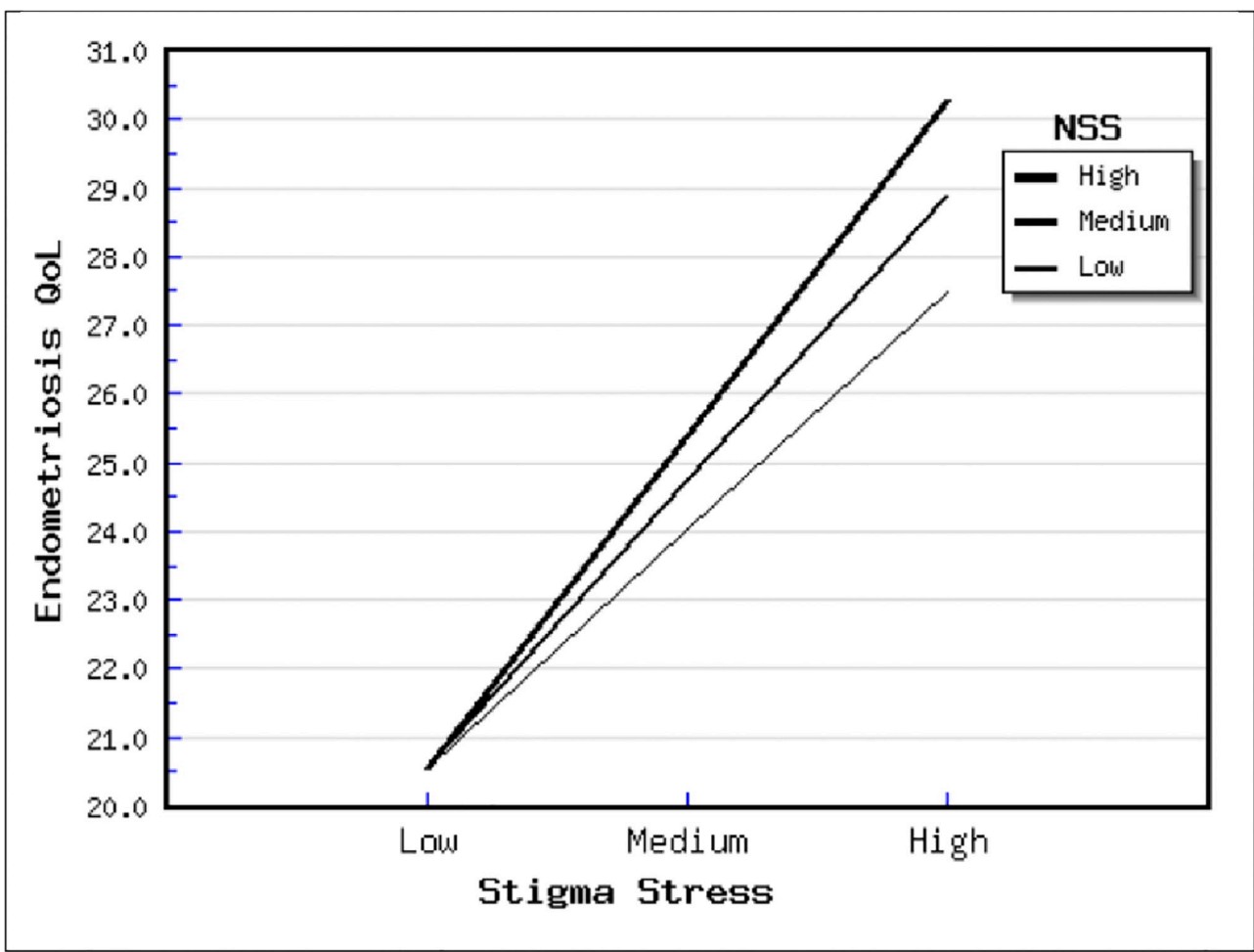

**Fig 2. Moderating effect of need for social support in the relationship between stigma stress and endometriosis QoL.**

Our findings identified high levels of endo-stigma in a sample of cisgender women from different Latin American and Caribbean countries, which is consonant with the existing research acknowledging stigma as a key psychosocial variable impacting the diagnosis and treatment of endometriosis in other cultural contexts [10, 14]. Moreover, our findings seem to support and expand previous research by providing much needed quantitative evidence of what previous qualitative literature had suggested, namely, that endo-stigma might be linked to worse endometriosis symptomatology and poor QoL [12]. Although previous literature has

**Table 6. Indirect effects hypotheses, results, conclusions, and mediation type.**

| Hypothesis (Med) | | IE | SE | t-value | p-value | CIBC | | Decision | Mediation (Type) |
|---|---|---|---|---|---|---|---|---|---|
| | | | | | | 2.5% | 97.5% | | |
| $H_{6a}$: | IP→SSE→EqoL | .083* | .028 | 2.963 | .003 | .035 | .146 | Supported | Yes (Complementary) |
| $H_{6b}$: | IP→SSE→SE | .073* | .031 | 2.332 | .020 | .021 | .142 | Supported | Indirect Effect Only |
| $H_{7b}$: | IP→SSE→SS | .135* | .042 | 3.211 | .001 | .051 | .213 | Supported | Indirect Effect Only |

Note:

*Significant; IP = Incapacitating Pain, SSE = Stigma Scale for Endometriosis; EqoL = Endometriosis QoL, SE = Self-Esteem, SS = Stigma Stress, IE = Indirect Effect, SE = Standard Error, CIBC = Confidence Interval Bias Corrected.

examined the association of endometriosis symptomatology in self-esteem and women's iden-tity [39, 40], our findings shows the key role of endo-stigma as a variable directly associated with self-esteem.

Another interesting finding was that stigma stress did not have a significant effect on endo-metriosis QoL and self-esteem in our study sample. This contrasts with previous literature sug-gesting that stigma could increase perceived stress, impacting patients' health and endometriosis QoL by exacerbating disease progression and symptomatology [13]. One poten-tial explanation might be related to specific social and cultural characteristics that impact the way stress is experienced as well as the stress scoping strategies employed. These findings high-light the need for more research on stigma stress, but also the role of specific social, cultural and structural variables unique to each context that might foster or buffer the experience of stigma stress among individuals living with endometriosis.

Finally, two additional findings merit special attention. Firstly, the results suggesting that those with high needs for social support also experience worse endometriosis QoL and self-esteem seems to be consistent with previous literature in other contexts. Specifically, qualita-tive literature has pointed the difficulties in interpersonal relationships were experiences stigmatization, lack of support and understanding from relatives, partners, and health profes-sionals [12, 41]. Thus, social support seems to be a consistently important variable in need of further research, particularly in the context of Latin America and the Caribbean. Secondly, the finding that endo-stigma partially mediated the relationship between incapacitating pain, endometriosis QoL and self-esteem is particularly important. This suggests that endo-stigma can be understood as a mechanism by which incapacitating pain could have even more detri-mental effects in the QoL and self-esteem of individuals living with endometriosis in Latin America and the Caribbean. This further highlights that, if public health intervention efforts are to be successful, they need to address endo-stigma as a fundamental variable in order to improve the physical and emotional wellbeing of patients.

## Limitations

This study has some limitations. Data is not representative of the entire Latin America and the Caribbean population due to its limited small sample size, non-probability sampling strategy, and the cross-sectional design of the study. Also, although we examined endo-stigma among patients from different Latin American and Caribbean countries, we did not examine specific variables related to the unique cultural context (i.e., public health policies, access to centralized and integrated care). Thus, our study did not explore potential similarities or differences across the different countries in the Latin American and Caribbean regions.

## Conclusion

Despite the prevalence of endometriosis worldwide and the potential negative consequences of endo-stigma, quantitative research efforts addressing it are scarce, particularly among under-represented and historically excluded contexts such as Latin America and the Caribbean. This study addresses that gap by examining endo-stigma and its role on the self-esteem and QoL of women living with endometriosis in Latin America and the Caribbean. Our findings suggest that stigma mediates the relationship between incapacitating pain and self-esteem. In addition, social support appears to be a key factor that moderates stigma stress and self-esteem in this sample. In sum, endo-stigma seems to be a key mechanism fundamental in for understanding and addressing the health and wellbeing of women with endometriosis in Latin America and the Caribbean. These results highlight the need for further research efforts addressing stigma as a key component of public health interventions.

## Supporting information

**S1 Text. Inclusivity questionnaire.**
(PDF)

## Acknowledgments

We thank the following patient associations around Latin America and the Caribbean for their support to this study: *Fundación Puertorriqueña de Pacientes con Endometriosis* (ENDOPR), *Asociación Colombiana de Endometriosis e Infertilidad (ASOCOEN)*, *Asociación Endometriosis Panamá (AENPA)*, *Endometriosis República Dominicana*, *Endometriosis México*, *Asociación Costarricense Endometriosis (AENDOCR)* and, *Endo Team Perú*.

## Author Contributions

**Conceptualization:** Ernesto Rosario-Hernández, Idhaliz Flores-Caldera, Eliut Rivera-Segarra.

**Formal analysis:** Ernesto Rosario-Hernández.

**Funding acquisition:** Eliut Rivera-Segarra.

**Investigation:** Yatzmeli Matías-González, Astrid Sánchez-Galarza.

**Methodology:** Ernesto Rosario-Hernández, Idhaliz Flores-Caldera.

**Writing – original draft:** Yatzmeli Matías-González, Astrid Sánchez-Galarza, Ernesto Rosario-Hernández, Idhaliz Flores-Caldera, Eliut Rivera-Segarra.

**Writing – review & editing:** Yatzmeli Matías-González, Astrid Sánchez-Galarza, Ernesto Rosario-Hernández, Idhaliz Flores-Caldera, Eliut Rivera-Segarra.

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
