## [Decision Letter · Decision Letter 0]

26 Jul 2022

PGPH-D-22-00601

Stigma and social support and their impact on quality of life and self-esteem among women with Endometriosis in Latin-America and the Caribbean

Dear Dr. Eliut Rivera-Segarra,

Thank you for submitting your manuscript to PLOS Global Public Health. After careful consideration, I feel that it has merit but does not fully meet PLOS Global Public Health’s publication criteria as it currently stands. Therefore, I invite you to submit a revised version of the manuscript that addresses the points raised during the review process by all the reviewers, particularly reviewer #1.

We look forward to receiving your revised manuscript.

Kind regards,

Rakesh Singh

Academic Editor

Journal Requirements:

3. Please send a completed 'Competing Interests' statement, including any COIs declared by your co-authors. If you have no competing interests to declare, please state "The authors have declared that no competing interests exist". Otherwise please declare all competing interests beginning with the statement "I have read the journal's policy and the authors of this manuscript have the following competing interests:"

4. Please provide separate figure files in .tif or .eps format and removed from the manuscript file.

5. In the online submission form, you indicated that "The manuscript presents the data used in this study. Dataset requests will be evaluated by the research team and shared upon reasonable request.". All PLOS journals now require all data underlying the findings described in their manuscript to be freely available to other researchers, either 1. In a public repository, 2. Within the manuscript itself, or 3. Uploaded as supplementary information.

Reviewers' comments:

Reviewer #1: The article addresses an understudied health condition that is disabling physically and psychologically.

Impact on many women worldwide. The article focuses on understanding the role of disease-related stigma on the quality of life among people affected by a chronic condition, which is the relevant angle that could inform policies and improve the quality of life of people affected by endometriosis. Despite the limitations, the study provides some helpful information for future studies.

Taking this into account, I recommend that the authors review the article and provide more information regarding the adaptation process of the questionnaires (beyond the statistical analysis). For example, it is unclear if the questionnaires were translated to Spanish, and if that was the case, how was it done.

On the sociodemographic data, it will be essential to know the rationale behind the inclusion of the variables (for example, why religion was asked) as well as the exclusion of other variables such as socioeconomic status or if the informants were part of an association/support group, as well as employment status.

Besides, while the authors recognised that they had not included health interventions among the variables analysed, a brief description of global, regional, and country-specific health programs/interventions/efforts to provide care to women living with endometriosis could allow to understand the study context.

Reviewer #2: Thank you for an interesting and necessary paper on an understudied condition in this region. I found a few typos in the abstract. The abstract also lists endo-stigma without having explained that term previously - please write the full term and then the abbreviation.

---

## [Editor Report · Decision Letter 1]

7 Nov 2022

Stigma and social support and their impact on quality of life and self-esteem among women with Endometriosis in Latin-America and the Caribbean

PGPH-D-22-00601R1

Dear Dr. Rivera-Segarra,

We are pleased to inform you that your manuscript 'Stigma and social support and their impact on quality of life and self-esteem among women with Endometriosis in Latin-America and the Caribbean' has been provisionally accepted for publication in PLOS Global Public Health.

Best regards,

Rakesh Singh

Academic Editor
